# Adiponectin Alleviates Cell Injury due to Cerebrospinal Fluid from Multiple Sclerosis Patients by Inhibiting Oxidative Stress and Proinflammatory Response

**DOI:** 10.3390/biomedicines11061692

**Published:** 2023-06-12

**Authors:** Marta Mallardo, Elisabetta Signoriello, Giacomo Lus, Aurora Daniele, Ersilia Nigro

**Affiliations:** 1CEINGE Biotecnologie Avanzate Franco Salvatore, 80145 Naples, Italy; marta.mallardo@unicampania.it (M.M.); ersilia.nigro@unicampania.it (E.N.); 2Dipartimento di Scienze e Tecnologie Ambientali, Biologiche, Farmaceutiche, Università della Campania “Luigi Vanvitelli”, 81100 Caserta, Italy; 3Centro di Sclerosi Multipla, II Clinica Neurologica, Università della Campania “Luigi Vanvitelli”, Via S. Pansini 5, 80131 Naples, Italy; elisabetta.signoriello@unicampania.it (E.S.); giacomo.lus@unicampania.it (G.L.); 4Dipartimento di Medicina Molecolare e Biotecnologie Mediche, “Federico II” Università degli Studi di Napoli, 80131 Naples, Italy

**Keywords:** adiponectin, multiple sclerosis, glial and neuronal cells, oxidative stress, inflammation

## Abstract

Multiple sclerosis (MS) is the most common disabling neurological disease characterized by chronic inflammation and neuronal cell viability impairment. Based on previous studies reporting that adiponectin exhibits neuroprotective effects in some models of neurodegenerative diseases, we analyzed the effects of AdipoRon treatment, alone or in combination with the cerebrospinal fluid of patients with MS (MS-CSF), to verify whether this adipokine acts on the basal neuronal cellular processes. To this aim, SH-SY5Y and U-87 cells (models of neuronal and glial cells, respectively) were exposed to MS-CSF alone or in co-treatment with AdipoRon. The cell viability was determined via MTT assay, and the possible underlying mechanisms were investigated via the alterations of oxidative stress and inflammation. MTT assay confirmed that AdipoRon alone did not affect the viability of both cell lines; whereas, when used in combination with MS-CSF, it reduces MS-CSF inhibitory effects on the viability of both SH-SY5Y and U-87 cell lines. In addition, MS-CSF treatment causes an increase in pro-inflammatory cytokines, whereas it determines the reduction in anti-inflammatory IL-10. Interestingly, the co-administration of AdipoRon counteracts the MS-CSF-induced production of pro-inflammatory cytokines, whereas it determines an enhancement of IL-10. In conclusion, our data suggest that AdipoRon counteracts the cytotoxic effects induced by MS-CSF on SH-SY5Y and U-87 cell lines and that one of the potential molecular underlying mechanisms might occur via reduction in oxidative stress and inflammation. Further in vivo and in vitro studies are essential to confirm whether adiponectin could be a neuro-protectant candidate against neuronal cell injury.

## 1. Introduction

Multiple sclerosis (MS) is a long-lasting (chronic) disease of the central nervous system (CNS) affecting approximately 2.5–3.5 million people worldwide [1,2]. Actually, the etiology of MS is incompletely known but seems to be due to a complex interplay between genetics and environmental factors [3]. In particular, the molecular causes underlying the myelin damage and oligodendrocytes death characteristics of MS are not yet fully understood [4]. Recently, there was evidence that an elevated nitric oxide (NO) amount and an increased expression of pro-inflammatory cytokines are associated with disease worsening [5], and accordingly, in vitro studies showed that NO and IFN-γ and TNF-α inhibit neuronal cells viability [6,7]. Beyond inflammation and oxidative stress, it was demonstrated that neuronal death is influenced also by additional factors, including excitatory toxicity, and mitochondrial malfunction, all processes regulated by sirtuins. The sirtuin family is a group of highly conserved nicotinamide adenine dinucleotide (NAD+)-dependent protein deacetylases constituted by seven species in mammals, sirtuin 1–7 (SIRT1-7) [8]. SIRT1 has a nuclear localization while SIRT3 has a mitochondrial one. Both SIRT1 and SIRT3 decrease microglia activation and inflammatory responses exerting a great influence on the regulation of oxidative stress and mitochondrial metabolism [9]. Studies on post-mortem human brain tissues showed reduced levels of SIRT3 expression in MS affected brains compared to control samples [10].

Adipose tissue is known to regulate biological processes such as energy balance and insulin sensitization through the secretion of bioactive molecules, known as adipokines [11,12]. Beyond energy balance regulation, a growing number of studies highlighted the importance of adipose tissue in the regulation of inflammatory and immune responses [13]. Such a plethora of functions is mediated by several mechanisms including adipokines that function as hormones mediating the crass talk between adipose tissue, immune system, and peripheral organs. However, it is important to notice that additional mechanisms cooperate with adipokines in regulating inflammatory responses; in addition to adipocytes, there are complex and hetereogenous resident cell populations such as fibroblasts, macrophages, and vascular constituents (lymphocytes, dendritic cells, etc.), all participating to the bidirectional cross talk between adipose and peripheral tissues [14].

Among the adipokines, adiponectin is the most abundant adipokine exclusively synthesized and secreted by adipocytes with beneficial effects on a wide range of physio-pathological processes [15]. Indeed, while the serum expression of adiponectin is inversely related to metabolic disorders, adiponectin levels appear to be upregulated in different immune diseases [16,17,18], and in some cases, strongly related to a bad prognosis [19]. Interestingly, in a previous study, we found that adiponectin levels in both serum and cerebrospinal fluid (CSF) are increased in MS patients compared to healthy controls and significantly correlated with higher activity of the disease and worse prognosis, suggesting a pivotal role of adiponectin in MS immune responses [20]. On the other hand, Keyhanian et al. found that higher levels of adiponectin were associated with a lower hazard of relapse [21]. Other published data outlined that obesity and being overweight influence the risk and severity of the disease, further supporting that adipose tissue is implicated in the pathogenesis of MS [22,23]. Thus, based on the above considerations, adiponectin appears to be an important molecular mediator in MS, although the molecular mechanisms underlying its activity remain to be determined.

In this scenario, and considering that, to our knowledge, no data are available on the effects of a combined treatment with adiponectin and CSF derived from MS patients (MS-CSF), the aim of the present study was to investigate the possible interring mechanism of adiponectin against MS-CSF toxicity.

We evaluated the in vitro effects of AdipoRon, a small-molecule agonist analog of endogenous adiponectin, on U87 and SH-SY5Y cells, models of glial and neuronal cells, respectively, alone or in association with CSF from MS patients. In detail, we evaluated whether AdipoRon interferes with the effects induced by MS-CSF in terms of vaibility and oxidative stress. In addition, we explored whether the co-treatments with MS-CSF and AdipoRon are associated with changes in the expression of the mRNA levels of INF-γ, TNF-α, and IL-10 that are the major cytokines involved in the altered MS immune response.

## 2. Materials and Methods

### 2.1. Patient’s

A total of 20 MS patients (9 females and 11 males) were recruited from the Multiple Sclerosis Center, Second Division of Neurology of University of Campania “Luigi Vanvitelli” at the moment of diagnosis, and CSF was collected before starting any treatment. In addition, patients were observed prospectively over time (4.5 years), as previously reported [3]. The demographic and clinical characteristics and body mass index (BMI) of all participants were recorded according to the clinical practice; for each patient, we measured the expanded disability status scale (EDSS) and total annualized relapse rate (ARR); in addition, brain and spinal cord magnetic resonance imaging (MRI), lumbar puncture with oligoclonal band evaluation, extensive autoimmune panel, and metabolic evaluation were also performed. The progression index (PI) and multiple sclerosis severity score (MSSS) were used to calculate MS severity. As shown in Table 1, we collected CSF from 10 relapsing–remitting (RRMS) and 10 progressive (PMS) patients. The two groups did not differ in age, sex, BMI, PI, oligoclonal bands, serum IgG, serum adiponectin, and CSF adiponectin; ARR, EDSS, MSSS, disease duration, EDSS at the end of follow up, and CSF IgG were associated to the disease course, and therefore worse values were found in RRMS patients (Table 1).

For the in vitro experiments, pooled samples from both RRMS and PMS were used.

The 5 control subjects suffering from migrane were recruited and CSF was collected; biochemical and clinical parameters are shown in Table 1.

### 2.2. Cell Culture

Human glioblastoma (U-87) and neuroblastoma (SH-SY5Y) cell lines were provided from the Culture Cell Lines Facility CEINGE Biotecnologie Avanzate Franco Salvatore, Napoli, Italy. Both cell lines were cultured in Dulbecco’s modified Eagle’s medium (DMEM) (Thermo Fischer Scientific, Waltham, MA, USA), supplemented with 10% fetal bovine serum (FBS) (Lonza, Basel, Switzerland), 1% L-glutamine (Sigma-Aldrich, St. Louis, MO, USA), 1% penicillin/streptomycin (Thermo Fischer Scientific, MA, USA) at 37 °C in a humidified atmosphere of 5% CO_2_. For all treatments, the medium was removed and the cells treated with increasing concentrations of AdipoRon (2.5, 5, and 10 μg/mL) and/or MS-CSF (10%) in DMEM were supplemented with 5% FBS for the different incubation times.

### 2.3. MTT Cell Viability Assay

Cell viability was determined by 3-[4.5-dimethylthiazol-2-yl]-2.5-dipheniltetrazolium bromide (MTT) colorimetric assay according to the manufacturer’s instructions. Briefly, U-87 and SH-SY5Y cells were seeded in 96-well plates (2 × 10^3^/well) and incubated overnight in DMEM 10% FBS medium. The day after, the cells were treated with increasing doses of AdipoRon (2.5, 5, and 10 μg/mL) (Sigma-Aldrich, MO, USA), or treated with MS-CSF from MS patients (10%) or co-treated with AdipoRon (2.5 μg/mL) and MS-CSF (10%). As control, SH-SY5Y and U-87 cells were incubated in 5% FBS medium alone. After 24, 48, and 72 h of treatment, cells were stained with MTT reagent and processed as previously described [24]. Each experiment was performed two times in triplicate for MS-CSF and AdipoRon treatment alone and three times for the combined treatments.

### 2.4. LDH Release Assay

The lactate dehydrogenase (LDH) assay was used for the quantification of cell death and cell lysis.

U-87 and SH-SY5Y cell lines were seeded in 96-well plates at a density of 2 × 104 cells/well and incubated overnight. The day after, the cells were treated with different doses of AdipoRon (2.5, 5, and 10 μg/mL) or treated with MS-CSF (10% *v*/*v*), or co-treated with AdipoRon (2.5 μg/mL) and MS MS-CSF (10% *v*/*v*). After 24, 48, and 72 h of exposure, each supernatant (100 μL) was treated with 100 μL of the reaction mixture (0.7 mM INT; 54 mM lactic acid; 0.3 mM phenazine methosulfate; and 0.8 mM NAD+ in 0.2 M Tris-HCl pH 8.0). The reaction was carried out as previously described [25]. Each experiment was performed three times in triplicate.

### 2.5. NO Determination

Nitric oxide (NO) release in culture supernatants was measured by the Griess reagent according to the manufacturer’s instruction. Briefly, U-87 and SH-SY5Y cells were treated with AdipoRon (2.5 μg/mL) or treated with MS-CSF (10% *v*/*v*), or co-treated with AdipoRon (2.5 μg/mL) and MS-CSF (10% *v*/*v*) for 24 h. The day after, supernatants (100 μL) were mixed with 50 μL of Griess reagent solution A (1% sulfanilamide *w*/*v* in 10% HCl 37% *v*/*v*) for 5 min and successively with 50 μL of Griess reagent solution B (0.1% N-(1-naphthyl) ethylenediaminedihydrochloride). After 10 min of incubation at room temperature, the absorbance was measured at 540 nm and referred to the absorbance of NaNO2 standard solutions, treated in the same way with the Griess reagents, by means of a calibration curve. Each experiment was performed two times in triplicate.

### 2.6. RNA Extraction and Quantitative Real Time-PCR

Total RNA was extracted from U-87 and SH-SY5Y cells by using TRIzol Reagent (Thermo Fischer Scientific, MA, USA). RNA concentration was quantified through the Qubit 4 Fluorometer (Thermo Fischer Scientific, MA, USA).

Successively, total RNA was subjected to reverse transcription with SuperScript III First-Strand Synthesis SuperMix (Thermo Fischer Scientific, Massachusetts, USA) according to the manufacturer’s instructions. Next, qPCR was performed using iQ SYBR Green Supermix (Bio-Rad, Hercules, CA, USA) in C1000 Touch Thermal Cycler (Bio-Rad, CA, USA). GAPDH was used as a housekeeping gene; fold changes were calculated with the 2^−ΔΔCt^ method (primer sequences and PCR reaction condition are available on request). Each experiment was performed three times in triplicate.

### 2.7. Statistical Analysis

Data are expressed as mean of replicates ± standard error of the mean (SEM). GraphPad Prism software-Update 6 (La Jolla, CA, USA) was used to carry out the analyses. The *p* values were determined by the one-way or two-way ANOVA followed by the Tukey or Dunnet multiple comparisons test was performed. A *p* value < 0.05 was considered statistically significant.

## 3. Results

### 3.1. CSF from MS Patients Reduces Neuronal and Glial Cell Viability

The effects of CSF from RRMS and PMS patients on glioblastoma and neuroblastoma cells in terms of viability was firstly evaluated. To this aim, both cell lines were treated with 10% MS-CSF in DMEM 5% FBS for 24, 48, and 72 h, and cell viability was assessed using the MTT assay. We could not find any difference in the two subgroups treatment, therefore we decided to consider the pooled CSF throughout the manuscript.

As shown in Figure 1, MS-CSF reduced the cell viability of SH-SY5Y (a) and U-87 (b) cell lines during all incubation times tested already after 24 h of incubation and persisting after 48 and 72 h.

As controls, we analyzed the effects of CSF from non-MS subjects on the viability of the same cell lines (Figure 1). These control-CSFs did not exert any relevant modification to cell viability compared to untreated cells both in SH-SY5Y (a) and U-87 (b).

### 3.2. Effects of AdipoRon Treatment on the Viability SH-SY5Y and U-87 Cells

The effects of AdipoRon on SH-SY5Y and U-87 cell viability were tested; both cell lines were treated in 5% FBS with different doses of AdipoRon (2.5, 5, and 10 μg/mL) for 24, 48, and 72 h, and cell viability was evaluated using MTT assay. As shown in Figure 2, the lowest dose of 2.5 μg/mL did not reduce cell viability at the incubation times tested. In addition, both cell lines’ viability was not influenced by AdipoRon administration after 24 h of treatment at any of the doses tested. At longer exposure times (48 and 72 h), the viability of both cell lines was negatively influenced by AdipoRon in a dose-dependent manner. In detail, the highest dose of 10 μg/mL showed a significant impact on the cell viability of both cell lines, while at the AdipoRon dose of 5 μg/mL, cell viability was significantly reduced only for U-87 cells.

Since the dose of 2.5 μg/mL of AdipoRon did not show toxic effects, the next experiments were performed using this dose.

### 3.3. AdipoRon Ameliorates the Cytotoxic Effects on Cell Viability Induced by MS-CSF

Previous studies reported that adiponectin exhibits neuroprotective effects in some models of neurodegenerative diseases [25]; therefore, we analyzed the effects of AdipoRon treatment, alone or in combination with MS-CSF, to elucidate its potential in interfering with MS-CSF-induced loss of cell viability. As shown in Figure 3, MTT assay confirmed that AdipoRon alone did not affect the viability of both cell lines, whereas, when used in combination with MS-CSF, it reduced MS-CSF inhibitory effects on the viability of both SH-SY5Y (a) and U-87 (c) cell lines.

A similar response was observed performing the LDH assay. Indeed, as shown in Figure 4, exposure to MS-CSF increased LDH release from both SH-SY5Y (b) and U-87 (d), whereas cells treated with the combination of AdipoRon and MS-CSF resulted in lower LDH release, suggesting that AdipoRon attenuates the increased cellular mortality induced by MS-CSF. In detail, AdipoRon is already effective in attenuating cell lysis after 24 h of incubation in SH-SY5Y cells, persisting after 48 and 72 h, whereas on U-87, AdipoRon significantly attenuates MS-CSF effects only at 24 h of treatment, albeit a less evident effect is also present after 48 and 72 h.

### 3.4. AdipoRon Ameliorates MS-CSF-Induced Nitric Oxide Release

Accumulating evidence suggests that nitric oxide (NO) plays a role in the immunopathogenesis of MS. In particular, an increased NO production was implicated as a mediator of demyelination and axonal damage in both MS and its animal models [26].

In this scenario, we tested the NO production, SIRT3 expression, and SOD2 expression in response to 10% MS-CSF and 2.5 μg/mL AdipoRon treatment, alone or in combination, for 48 h in SH-SY5Y (a–c) and U-87 neuronal cells (d–f). We found that MS-CSF from MS patients significantly increased NO production in both cell lines tested, whereas combined treatment with AdipoRon attenuates the pro-NO effects of MS-CSF in both cell lines. The administration of AdipoRon alone is unable to induce the release of NO in both cell models.

Regarding SIRT3, CSF exposure significantly reduces SIRT3 expresion in U87 cells © and in SH-SY5Y (b), although not significantly. Importantly, in both cell lines, AdipoRon in combined treatment increased SIRT3 levels. MS-CSF exposure significantly reduces SOD2 expresion in SH-SY5Y (c) and in U87 cells (f), although not significantly; when in co-treatment, AdipoRon in combined treatment increased SOD2 levels in both cell lines. Altogether, such data indicate that AdipoRon reduce MS-CSF induction of oxidative stress.

### 3.5. AdipoRon Alleviates the MS-CSF Effects on the mRNA Levels of TNF-α, INF-γ, and IL-10

To determine whether the effects on cell viability induced by MS-CSF were also associated with changes in the expression of the inflammatory cytokines, we evaluated in both cell lines the mRNA levels of INF-γ, TNF-α, and IL-10, the main cytokines involved in MS immune response, after treatment for 48 h with MS-CSF (10% *v*/*v*) and AdipoRon (2.5 μg/mL) alone or in combination. As shown in Figure 5, the results of the qPCR show that MS-CSF treatment promotes a significant increase in pro-inflammatory INF-γ and TNF-α mRNA levels and a decrease in that of anti-inflammatory IL-10, compared to the untreated control. In detail, INF-γ and TNF-α were significantly enhanced by MS-CSF treatment in both SH-SY5Y (Figure 5a,b) and in U-87 (Figure 5d,e), whereas MS-CSF administration strongly reduced the levels of IL-10 (Figure 5c,f). AdipoRon alone did not affect the expression of the tested cytokines. Interestingly, AdipoRon co-administration significantly attenuated the MS-CSF-inducing effects on INF-γ and TNF-α and the reduction in IL-10 in SH-SY5Y (Figure 5a–c) and U-87 cells (Figure 5d–f).

## 4. Discussion

In this study, we investigated the molecular mechanisms underlying adiponectin in MS, finding that the administration of AdipoRon (a small agonist analog of adiponectin) improves neuronal and glial survival and alleviates the viability inhibition that induces MS-CSF administration. In detail, we confirmed the in vitro toxicity induced by MS-CSF treatment [27] and observed that AdipoRon ameliorated viability and survival of both cell types damaged by the exposure to MS-CSF. Additionally, interestingly, we found that AdipoRon partially reverts such effects through modulation of oxidative stress and inflammation.

A growing number of studies highlighted the importance of adipose tissue in immune and inflammatory disorders mainly through the secretion of adipokines [28,29]. Among the adipokines, adiponectin plays a significant role in several autoimmune diseases, including MS [30,31].

Recently, we and others [20,28,32] reported that an increased expression of adiponectin in serum and in the CSF in patients affected by MS might potentially be considered as a prognostic biomarker in MS. Functionally, adiponectin could contribute to a chronic inflammatory state linked to a worse disease prognosis as suggested by the correlation of adiponectin with some parameters: higher PI, MSSS, and higher baseline EDSS score [32].

In this scenario, we aimed to investigate some molecular mechanisms underlying adiponectin action in MS, finding that the administration of AdipoRon can facilitate neuronal and glial survival by the recovery of MS-CSF-induced viability inhibition. In detail, adiponectin ameliorated the viability of both cell types damaged by the exposure to MS-CSF. We confirmed the in vitro toxicity induced by MS-CSF as previously demonstrated [27], although the unknown protein and metabolite profile of MS-CSF prevents the recognition of the factors driving these effects.

Interestingly, we found that adiponectin partially reverts such effect through modulation of oxidative stress and inflammation. Neuronal cells are highly sensitive to oxidative and nitrosative stress due to their high oxygen request [33]. The overproduction of ROS and NO alters brain function, as observed in many neurodegenerative diseases, leading to neuronal cell death [34]. In particular, the concentrations of nitrate and nitrite, markers of NO production, are higher in the MS-CSF compared to control subjects and could induce disruption of the blood–brain barrier, oligodendrocyte injury, demyelination, and axonal degeneration, culminating in cell death [35]. In our study, we found an increase in NO generation in cells under the presence of MS-CSF that could at least in part explain the inhibition of observed neuronal viability. It is interesting to notice that AdipoRon administration was able to counteract MS-CSF effects in the co-treatment condition, strongly suggesting that AdipoRon can promote neuronal survival, at least in part, through the reduction in nitrosative stress.

The main pathological trademarks of MS are cellular and myelin damage, mainly due to an altered immune response that results in the production of a large amount of pro-inflammatory cytokines by CD4+ T cells [36]. Our results suggest that MS-CSF treatment caused an increase in pro-inflammatory cytokines levels, whereas it caused the reduction in anti-inflammatory IL-10. Interestingly, we found that the AdipoRon administration can counteract the MS-CSF-induced production of pro-inflammatory cytokines, while enhancing IL-10 levels. In line with our results, it was shown that adiponectin can inhibit Th17 cell-induced pro-inflammatory cytokine cascade, which is typical of MS [37].

Considering these results, the role of adiponectin in MS could be related to the counteraction of the oxidative stress and chronic inflammation, two hallmarks of MS. Accordingly, the in vivo study published by Piccio et al. showed that adiponectin knockout mice with experimental autoimmune encephalomyelitis (EAE) develop a more severe phenotype, and that the administration of external adiponectin ameliorates their condition [38].

In support of our hypothesis of the counteraction of the inflammatory state by adiponectin, Zhang et al. demonstrated that adiponectin can inhibit Th17 cell-mediated central nervous system inflammation [37]. In addition, several in-vitro studies on many other cell types and models outline the potential anti-inflammatory properties of adiponectin [39].

The main limitation of the study is the use of tumor cells in place of primary neurons and glial cells; future studies using primary cultured cells would confirm our data and deepen the molecular mechanisms at the basis of CSF and AdipoRon actions. Another limitation of the study is related to the relative low number of recruited patients and CSF; although, considering the awkwardness of recruiting patients, future studies with larger sample sizes will improve knowledge in this field. This potentially impacted also the absence of any relevant difference in the effects induced by the MS-CSF collected by RRMS or PMS patients.

In conclusion, our findings provide a further step towards the understanding of the involvement of adiponectin in MS, demonstrating that adiponectin protects neuronal and glial cells from cellular stress induced by CSF derived from patients affected by MS. Further studies in co-cultures of neuronal and glial cells [40] or heterotypic brain precision slice cultures are needed to clarify the contribution of adiponectin to the complex cytokine network working in MS.

Nonetheless, altogether, our data outline the role of adiponectin in linking adipose tissue with MS.

## Figures and Tables

**Figure 1 biomedicines-11-01692-f001:**
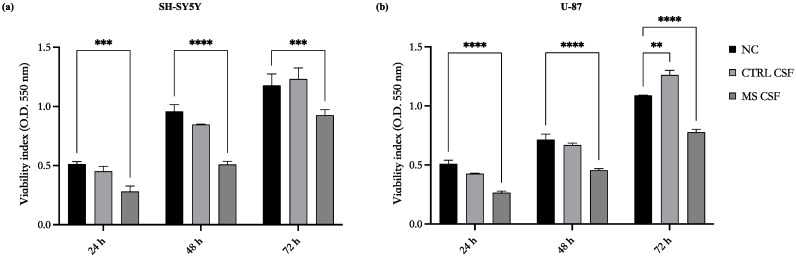
Cerebrospinal fluid from MS patients (MS-CSF) reduces the viability of SH-SY5Y (**a**) and U-87 cells (**b**) while CSF from controls did not produce any effect. Cell viability was assessed by MTT assay after exposure for 24, 48, and 72 h with 10% MS-CSF from MS patients and control subjects. Values are expressed as the mean of two different experiments performed in triplicate ± standard error of the mean (SEM). The statistical analysis was evaluated using the two-way ANOVA test. ** *p* value < 0.05; *** *p* value < 0.001; **** *p* value < 0.0001.

**Figure 2 biomedicines-11-01692-f002:**
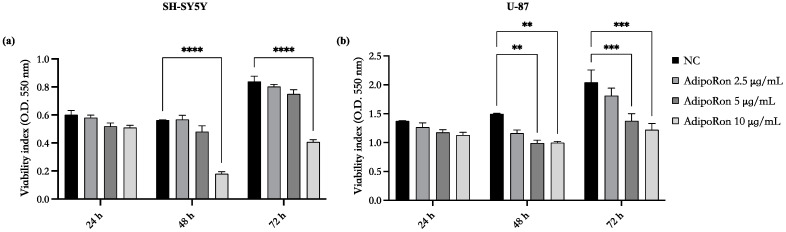
Effects of different doses of AdipoRon on the viability of SH-SY5Y (**a**) and U-87 cells (**b**). SH-SY5Y and U-87 cells were plated in a 96-well plate and incubated with AdipoRon (2.5, 5, and 10 μg/mL). After 24, 48, and 72 h of incubation, cell viability was measured using MTT assay. Values are expressed as the mean of two different experiments (sample size n = 24) ± standard error of the mean (SEM). The statistical analysis was evaluated using the two-way ANOVA test. ** *p* value < 0.01; *** *p* value < 0.001; and **** *p* value < 0.0001.

**Figure 3 biomedicines-11-01692-f003:**
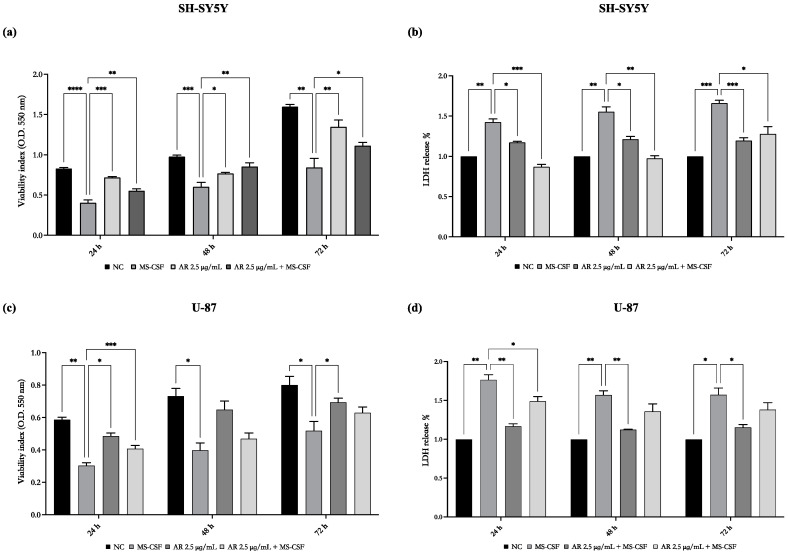
AdipoRon partially attenuates the cytotoxic effects induced by MS-CSF on the viability of SH-SY5Y and U-87 cells. The cells were treated with MS-CSF and AdipoRon alone or in combination for 24, 48, and 72 h; successively, cell viability was evaluated by both MTT (**a**,**b**) as well as LDH assay (**c**,**d**). Values are expressed as a mean of three different experiments (sample size = 36) ± standard error of the mean (SEM). The statistical analysis was evaluated using the two-way ANOVA test. * *p* value < 0.05; ** *p* value < 0.01; *** *p* value < 0.001; and **** *p* value < 0.0001 versus MS-CSF.

**Figure 4 biomedicines-11-01692-f004:**
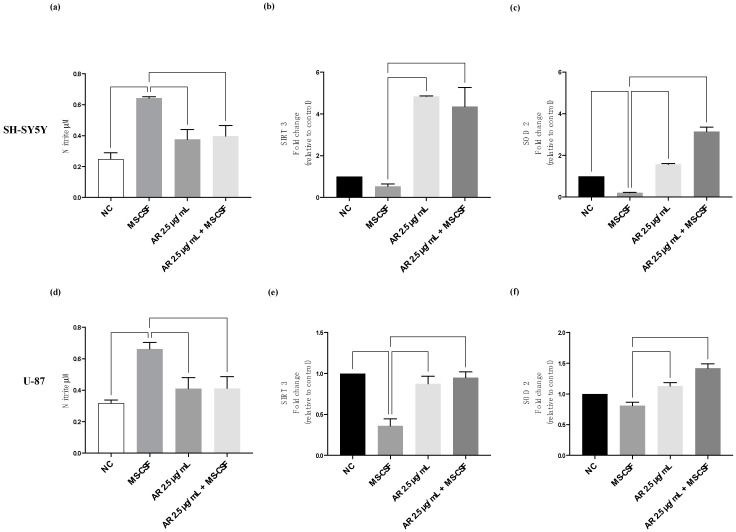
Neuroprotective effects of AdipoRon on MS-CSF-induced nitrosative stress. NO, SIRT3, and SOD2 levels in SH-SY5Y and U87 cells were measured. MS-CSF exposure significantly promoted NO production in SH-SY5Y (**a**) and U87 (**d**) cells. AdipoRon in combined treatment decreased MS-CSF-induced NO levels in both cell lines. MS-CSF exposure significantly reduces SIRT3 expresion in U87 cel^©^ (**e**) and in SH-SY5Y (**b**). AdipoRon in combined treatment increased SIRT3 levels in both cell lines. MS-CSF exposure significantly reduces SOD2 expresion in SH-SY5Y (**c**) and in U87 cells (**f**). AdipoRon in combined treatment increased SOD2 levels in both cell lines. Values are expressed as the mean of two different experiments performed in triplicate ± standard error of the mean (SEM). The statistical analysis was evaluated using the one-way ANOVA test.

**Figure 5 biomedicines-11-01692-f005:**
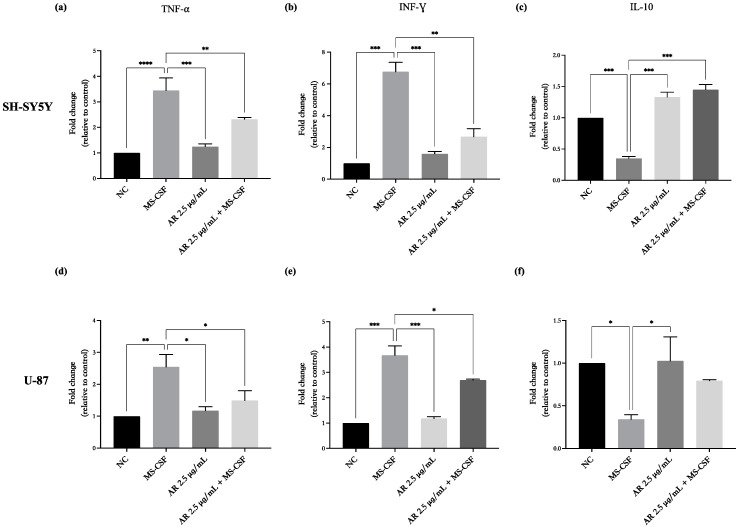
AdipoRon counteracts the MS-CSF effects on mRNA levels of TNF-α, INF-γ, and IL-10 on SH-SY5Y and U87 cells. SH-SY5Y and U87 cells were treated with MS-CSF (10%) and AdipoRon (2.5 μg/mL) alone or in combination for 48 h. MS-CSF treatment results in the increase in the expression of TNF-α (**a**,**d**), INF-γ (**b**,**e**), and reduces IL-10 (**c**,**f**) on both cell lines. AdipoRon strongly attenuated MS-CSF-induced effects on inflammatory cytokines in both cell lines. Untreated cells were used as negative control (NC). The experiment was performed three times in triplicate. The statistical analysis was evaluated using the one-way ANOVA test. * *p* value < 0.05; ** *p* value < 0.01; *** *p* value < 0.001; and **** *p* value < 0.0001 versus MS-CSF.

**Table 1 biomedicines-11-01692-t001:** Anthropometric, clinical and biochemical characteristic of MS patients.

	Control Subjects = 5	RRMS = 10	PMS = 10	*p* Value ^c^
Age	45.4 ± 20.90	43 ± 8.13	48 ± 11.36	0.28
Sex (M/F)	3/2	5/5	6/4	-
BMI	24 ± 3.4	24 ± 4.89	25 ± 2.28	0.60
CSF IgG (mg/dL)	3.07 ± 1.3	11.9 ± 8.4	3.7 ± 1.67	0.03
Oligoclonal bands	N/A	11 ± 8.6	8.63 ± 3.8	0.49
Serum IgG (mg/dL)	1250 ± 188.5	1248 ± 291	1114.8 ± 313.2	0.40
Serum adiponectin (μg/mL) ^a^	10.20 μg/mL	13.10 ± 2.17	12.60 ± 1.76	0.58
CSF adiponectin (ng/mL) ^b^	6.02 ± 2.74	15.15 ± 4.44	11.67 ± 5.51	0.18
Serum albumin (μg/mL)	4.19 ± 0.24	4.33 ± 0.13	3.88 ± 0.5	0.04
Total ARR	-	0.85 ± 0.51	0.21 ± 0.30	0.0035
PI	-	1.46 ± 2.09	1.01 ± 1.14	0.57
Basal EDSS	-	1.94 ± 2.17	3.7 ± 3.16	0.039
MSSS	-	3.77 ± 3.40	6.65 ± 1.99	0.039
Disease duration	-	12.1 ± 3.1	8.3 ± 1.6	0.003
EDSS at the end of follow up	-	2 ± 2.33	4.5 ± 0.97	0.0063
Follow-up (years)	-	8.8 ± 2.1	7.1 ± 0.65	0.03
Link index (CSF IgG/serum IgG × serum albumin/CSF albumin)	-	0.97 ± 0.71	0.71 ± 0.23	0.35
Barrier index (CSF albumin/serum albumin × 100)	-	0.68 ± 0.44	0.48±0.19	0.30
PCR (mg/dL)	-	0.13 ± 0.1	0.16 ± 0.14	0.61

^a,b^ Data reported from a previous study. ^c^ *p*-values referred to RRMS and PMS groups.

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
