# Peer review of "Adiponectin Alleviates Cell Injury due to Cerebrospinal Fluid from Multiple Sclerosis Patients by Inhibiting Oxidative Stress and Proinflammatory Response"

_biomedicines, 2023, doi:10.3390/biomedicines11061692_

Round 1

Reviewer 1 Report

Thank you very much for allowing me to review the article titled "Adiponectin alleviates cell injury due to cerebrospinal fluid from multiple sclerosis patients by inhibiting oxidative stress and inflammatory response" (biomedicines-2415840). This manuscript is submitted for the "Neurobiology and Neurologic Disease" section of the Special Issue "Cerebrospinal Fluid: The Potential for Molecular Disease-Related and Drug-Related Biomarkers" of Biomedicine.

It is always advisable not to use acronyms in the title, so I suggest replacing CSF. This is a prospective study whose aim was to investigate the in vitro effects of AdipoRon, a small-molecule agonist analog of endogenous adiponectin, on U87 and SH-SY5Y cells, models of glial and neuronal cells respectively, alone or in association with cerebrospinal fluid from MS patients. The second objective was to evaluate whether AdipoRon interferes with the effects induced by MS-CSF in terms of proliferation and oxidative stress. Additionally, we explored whether the co-treatments with MS-CSF and AdipoRon are associated with changes in the expression of mRNA levels of INF-γ, TNF-α, and IL-10, which are the major cytokines involved in the altered MS immune response.

The introduction should be expanded to further explain the physiopathological basis of adipose tissue in inflammatory-related disorders and the hypothesis regarding the effects of a combined treatment with adiponectin and MS-CSF derived from MS patients. This would allow for a better assessment of the contribution of this work. Since this study is a continuation of a previous study, it would be advisable to explain the differences a bit more.

In the results section, these results regarding the graphs should be further explained.

The main limitation of the study is the sample size; however, I believe that the results can be used to consider larger sample sizes in order to advance knowledge in this field.

Author Response

Reviewer 1

Thank you very much for allowing me to review the article titled "Adiponectin alleviates cell injury due to cerebrospinal fluid from multiple sclerosis patients by inhibiting oxidative stress and inflammatory response" (biomedicines-2415840). This manuscript is submitted for the "Neurobiology and Neurologic Disease" section of the Special Issue "Cerebrospinal Fluid: The Potential for Molecular Disease-Related and Drug-Related Biomarkers" of Biomedicine.

It is always advisable not to use acronyms in the title, so I suggest replacing CSF.

Reply: DONE

This is a prospective study whose aim was to investigate the in vitro effects of AdipoRon, a small-molecule agonist analog of endogenous adiponectin, on U87 and SH-SY5Y cells, models of glial and neuronal cells respectively, alone or in association with cerebrospinal fluid from MS patients. The second objective was to evaluate whether AdipoRon interferes with the effects induced by MS-CSF in terms of proliferation and oxidative stress. Additionally, we explored whether the co-treatments with MS-CSF and AdipoRon are associated with changes in the expression of mRNA levels of INF-γ, TNF-α, and IL-10, which are the major cytokines involved in the altered MS immune response.

The introduction should be expanded to further explain the physiopathological basis of adipose tissue in inflammatory-related disorders and the hypothesis regarding the effects of a combined treatment with adiponectin and MS-CSF derived from MS patients. This would allow for a better assessment of the contribution of this work. Since this study is a continuation of a previous study, it would be advisable to explain the differences a bit more.

Reply: We thank the reviewer for the suggestions and accordingly modified the introduction further elaborating on the pathophysiological role of adipose tissue in inflammatory disorders. We hope that now the hypothesis of our work is clearer. In the results section, these results regarding the graphs should be further explained.

Reply: accordingly, we have implemented the results section. In addition, to make clearer the comprehension for the readers, we also added additional experiments (see the results section):

  • In figure 1, we have included viability data on MS from control subjects.
  • In panels b, c, e, f of figure 4 we included data about the expression of SIRT3 and SOD2 levels, to better clarify and explain the regulation of oxidative stress processes by MS-CSF in combination or not with AdipoRon.

The main limitation of the study is the sample size; however, I believe that the results can be used to consider larger sample sizes in order to advance knowledge in this field.

Reply: we sincerely thank the reviewer for this observation and empathized in the discussion section this point (see lines 344-346).

Reviewer 2 Report

Multiple sclerosis (MS) is a chronic autoimmune disease that affects the central nervous system (CNS) and characterized by demyelination and axonal loss. In the present study, the authors investigated the toxic effect of the cerebrospinal fluid samples obtained from MS patients (MS-CSF) in human neuroblastoma cells and human glioblastoma cells. Because the authors previously found that adiponectin, which is known as an anti-inflammatory and antioxidative adipokine, they further investigated the protective effects of an adiponectin agonist AdipRon against MS-CSF. The topic of the present manuscript is interesting, however, the results shown here is preliminary and I hesitate to support the publication of the current form.

1. Throughout the manuscript, “NC” or “negative control” has been used as the control. If the “NC” means not-treated cells, data with non-MS control CSF samples must be included.

2. The current version has no implication regarding the toxic or pro-inflammatory mechanisms of MS-CSF. Thus, the readers will learn nothing about the toxic effects of MS-CSF and the protective effects of AdipoRon from the current version.

3. Neurons usually do not proliferate and the current data were obtained by using neuroblastoma cells which are tumor cells and not neurons. Thus, the statements regrading the proliferation of SH-SY5Y cells has no implication of the neuronal or axonal pathology in MS. As for the proinflammatory response of glioblastoma cells, it is not sure whether pro-inflammatory cytokines secreted by astrocytes contribute to axonal loss. MS is characterized by demyelination, why the authors did not use oligodendrocytes?

4. Lines 313-, the limitation of the study, I disagree with the authors. The limitation of the study is the use of tumor cells. Using primary cultured cells would increase the impact of this study.

Minor

On line 216, a reference is missing.

Author Response

Reviewer 2:

Multiple sclerosis (MS) is a chronic autoimmune disease that affects the central nervous system (CNS) and characterized by demyelination and axonal loss. In the present study, the authors investigated the toxic effect of the cerebrospinal fluid samples obtained from MS patients (MS-CSF) in human neuroblastoma cells and human glioblastoma cells. Because the authors previously found that adiponectin, which is known as an anti-inflammatory and antioxidative adipokine, they further investigated the protective effects of an adiponectin agonist AdipRon against MS-CSF. The topic of the present manuscript is interesting, however, the results shown here is preliminary and I hesitate to support the publication of the current form.

  1. Throughout the manuscript, “NC” or “negative control” has been used as the control. If the “NC” means not-treated cells, data with non-MS control CSF samples must be included.

Reply: we apologize for the lack of clarity.

The negative controls (NC) used here correspond to the respective cell lines that have not undergone any treatment. The referee is correct in identifying as negative controls those corresponding to CSF-treated cell lines from non-MS patients. Actually, during the first MTT test performed, we used CSF from subjects negative to the oligoclonal bands test: the preliminary data obtained clearly indicated that these CSFs do not cause toxic effects on cell lines in terms of viability. Therefore, we decided not to proceed for the subsequent experiments using CSF from individuals without MS (as a negative control) also due to the very limited number (5) of "control" patients at our disposal. In the revised version of the manuscript, we have included previous MTT data in Figure 1 (panels a, b) and anthropometrical/biochemical data of these control subjects in Table 1.

  1. The current version has no implication regarding the toxic or pro-inflammatory mechanisms of MS-CSF. Thus, the readers will learn nothing about the toxic effects of MS-CSF and the protective effects of AdipoRon from the current version.

Reply: We agree with the reviewer that we do not delve much into the molecular mechanisms of MS-CFS actions, but our data strongly suggest a modulation of cell viability, through oxidative stress and inflammation.

Then, the aim of the study was focused on the possible involvement of adiponectin in such molecular mechanisms. In particular, we investigated whether AdipoRon could have effects in modulating MS-CSF-induced toxicity in terms of viability, inflammatory and oxidative stress levels of neuronal and glial cells. Therefore, we first analyzed the effects of MS-CSF alone and later in combination with AdipoRon, observing that this molecule reduces the toxic effects due to MS-CSF in terms of cell viability and oxidative stress induction. We also investigated effects related to cytokine regulation.

In the revised version of the manuscript, we included additional data regarding the modulation of SOD2 and sirtuin 3 expression by MS-CSF and/or AdipoRon (see Figure 4 panels c, d e and f).

Indeed, sirtuins and SOD2 have been identified as a key regulator of processes affecting neuronal death, including inflammation, and oxidative stress. The modulation of the expression of both molecules further confirms that the effects that AdipoRon determines in counteracting the toxicity of MS-CSF and that they are at least in part mediated by the attenuation of oxidative stress.

  1. Neurons usually do not proliferate, and the current data were obtained by using neuroblastoma cells which are tumor cells and not neurons. Thus, the statements regarding the proliferation of SH-SY5Y cells has no implication of the neuronal or axonal pathology in MS. As for the proinflammatory response of glioblastoma cells, it is not sure whether pro-inflammatory cytokines secreted by astrocytes contribute to axonal loss. MS is characterized by demyelination, why the authors did not use oligodendrocytes?

Reply:  we totally agree with the reviewer, the use of cell lines is not the ideal experimental model to study the health and toxicity of neurons and glia. However, we didn't have all the tools and facilities to work with primary cells and we decided to produce a proof of concept, a pilot study that could perhaps pave the way for future research in the field of sclerosis multiple. In the current version, this has been clearly identified as the main limitation of the study in the Discussion section (lines 824-826).

  1. Lines 313-, the limitation of the study, I disagree with the authors. The limitation of the study is the use of tumor cells. Using primary cultured cells would increase the impact of this study.

Reply: we thank the reviewer for the suggestion and, accordingly, we have modified this section.

Minor

On line 216, a reference is missing.

Reply: we apologize for the mistake and thank the reviewer for the observation.

Reviewer 3 Report

Mallardo and colleagues provide in vitro evidence of the positive effects of adipo-Ron on MS-CSF induced neuronal and glial cell death. These initial experiments suggest partial resolution of both oxidative stress and inflammation in the mediation of adipo-Ron’s effects. More comprehensive studies are required to validate these findings. The manuscript can be readily improved based on some of the below suggestions.

Major comments:

What is the n per experimental group for your MTT, NO, and LDH experiments? For gene expression experiments, can you clarify if your sample size is n=3, or did you run triplicate experiments with an increased number of biological replicates?

Did you have the MS-CSF samples analyzed for substrate compositions (I.e., protein and metabolite profiling)? While it’s very interesting that the MS-CSF milieu is cytotoxic, recognizing which factors are largely driving these effects (beyond adiponectin R signaling pathways) would be immensely valuable? Distinguishing the CSF profile b/w the RRMS and PMS groups may also be very insightful.

For lines 204-206: I respectfully semi-disagree with the interpretation of the effects stated for the U-87 cells. While I recognize that viability is restored at both the 24 and 72 timepoints; to me, the effects, which are less robust than observed in SY5Y cells, are seen in U-87 cells at the 24-hour time point but generally do not appear to persist beyond.

The asterisks on figure 4 are missing.

For line 220: fix the concentration unit as it has @ instead of µl; section 3.5 Greek symbols are also the same issue (@ symbol in place of Greek symbol). 

Can you be consistent with your use of the terms ‘cell viability’, ‘cell growth’, ‘proliferation/replication’ (i.e., lines 65, 233, 260, 278). From a biological perspective, though related, these are discrete processes. Are you implying that adipo-Ron attenuated the MS-CSF stimulated decrease in cell replication, cell death, or cell growth?

To further support your findings, did you evaluate transcriptional differences in genes encoding oxidative stress proteins due to adipo-Ron treatment of MS-CSF treated? I suggest that if you still have cDNA or RNA, you run an oxidative stress panel.

Any evidence of the combined effects in co-cultures of U87 + SH-SY5Y or heterotypic brain precision slice cultures?

Glial U87 cells protect neuronal SH-SY5Y cells from indirect effect of radiation by reducing oxidative stress and apoptosis

academic.oup.com

You indicate that adiponectin is upregulated in the CSF (and serum) of MS patients. Can you further discuss the role of adiponectin receptor insensitivity in the mediation of MS. Based on the former, do you think that your MS-CSF protocol appropriately represents the MS ‘brain’?   Beyond cell viability, were there any other parameters that are characteristic of MS analyzed?

The effects of MS-CSF on cell viability using your cell model systems appear aggressive and immediate. Your cell data suggests that the adiponectin receptors are still sensitive to the effects of its agonists. In MS patients, where adiponectin is elevated but likely not signaling appropriately through its receptor, would Adipo-Ron have diminished efficacy? Also, can you discuss any conflicting reports where adiponectin is inversely related to MS severity.

As there have been reports of sex differences in RRMS risk, why was the data not analyzed based on sex? Were there any analyzed parameters that deviated due to sex condition? Additionally, adolescent obesity status influences risk (and severity) of MS. Were there any prior history of obesity/overweight indicated for MS patients used in this study?

Association between comorbidity and clinical characteristics of MS - PubMed

nlm.nih.gov

A blue and white logo

Description automatically generated with low confidence

Influence of body mass index on psychological and functional outcomes in patients with multiple sclerosis: a cross-sectional study - PubMed

nlm.nih.gov

A blue and white logo

Description automatically generated with low confidence

Minor comments:

There is inconsistency in figure legend labels and what you indicate in the text. In the results section, you indicate figure 4 A, B for SH-SY5Y and U-87, respectively; but this is not the order in the actual figure. Please double check all your figures and ensure they are consistent.

You are missing a citation on line 216.

Please check for grammatical errors and syntax (i.e., lines 195-196, lines 258 and 288). Some sentences contain weird wording (i.e., lines 260-261, 313-316).

The statement on lines 313-314 suggest increasing sample size to possibly see differences b/w RRMS and PMS patients. In the methods section, can you indicate that your MS-CSF is pooled from n=20 RRMS and PMS subjects.

For the title, consider ….inhibiting oxidative stress and proinflammatory  response.

Author Response

Reviewer 3

Mallardo and colleagues provide in vitro evidence of the positive effects of adipo-Ron on MS-CSF induced neuronal and glial cell death. These initial experiments suggest partial resolution of both oxidative stress and inflammation in the mediation of adipo-Ron’s effects. More comprehensive studies are required to validate these findings. The manuscript can be readily improved based on some of the below suggestions.

Major comments:

What is the n per experimental group for your MTT, NO, and LDH experiments? For gene expression experiments, can you clarify if your sample size is n=3, or did you run triplicate experiments with an increased number of biological replicates?

Reply: we apologize for the forgetfulness. Accordingly, we have modified the paper clarifying this point for each experiment and figure.

Did you have the MS-CSF samples analyzed for substrate compositions (I.e., protein and metabolite profiling)? While it’s very interesting that the MS-CSF milieu is cytotoxic, recognizing which factors are largely driving these effects (beyond adiponectin R signaling pathways) would be immensely valuable? Distinguishing the CSF profile b/w the RRMS and PMS groups may also be very insightful.

Reply: thanks to the reviewer for the interesting observation. Unfortunately, we do not have data on the profiling of CSF metabolites, which represents a field of great interest. furthermore, the small number and amount of samples makes planning such analysis quite difficult. However, in the revised version of the manuscript, we discuss this interesting point (see lines 791-792).

For lines 204-206: I respectfully semi-disagree with the interpretation of the effects stated for the U-87 cells. While I recognize that viability is restored at both the 24 and 72 timepoints; to me, the effects, which are less robust than observed in SY5Y cells, are seen in U-87 cells at the 24-hour time point but generally do not appear to persist beyond.

Reply: we sincerely apologize for the inaccuracy. Consequently, in the revised version of the manuscript, we corrected the mistake and clarified that the most noticeable effect occurs after 24 hours of treatment.

The asterisks on figure 4 are missing.

Reply: we apologize for the mistake. Accordingly, we modified the figure 4.

For line 220: fix the concentration unit as it has @ instead of µl; section 3.5 Greek symbols are also the same issue (@ symbol in place of Greek symbol). 

Reply: we apologize for the mistake. Accordingly, we modified.

Can you be consistent with your use of the terms ‘cell viability’, ‘cell growth’, ‘proliferation/replication’ (i.e., lines 65, 233, 260, 278). From a biological perspective, though related, these are discrete processes. Are you implying that adipo-Ron attenuated the MS-CSF stimulated decrease in cell replication, cell death, or cell growth?

Reply: we totally agree with the reviewer and apologize for the inaccuracy. Accordingly, we have modified the text to specify that AdipoRon attenuated the MS-CSF stimulated decrease in cell viability.

To further support your findings, did you evaluate transcriptional differences in genes encoding oxidative stress proteins due to adipo-Ron treatment of MS-CSF treated? I suggest that if you still have cDNA or RNA, you run an oxidative stress panel.

Reply: we thank the reviewer for the suggestion. Accordingly, we have added the evaluation of two more factors implicated in the oxidative stress: SIRT3 and SOD2. In the revised version of the manuscript, we included additional data regarding the modulation of SOD2 and sirtuin 3 expression by MS-CSF and/or AdipoRon (see Figure 4 panels c, d e and f).

Indeed, sirtuins and SOD2 have been identified as a key regulator of processes affecting neuronal death, including inflammation, and oxidative stress. The modulation of the expression of both molecules further confirms that the effects that AdipoRon determines in counteracting the toxicity of MS-CSF and that they are at least in part mediated by the attenuation of oxidative stress.

Any evidence of the combined effects in co-cultures of U87 + SH-SY5Y or heterotypic brain precision slice cultures?

Glial U87 cells protect neuronal SH-SY5Y cells from indirect effect of radiation by reducing oxidative stress and apoptosis

academic.oup.com

Reply: We thank the reviewer for the very interesting suggestion. Unfortunately, at present, we do not have data on U87+SH-SY5Y co-cultures or on heterotypic cultures of precision brain slices; this could represent the next step towards understanding the actions of adiponectin in a more complex cellular system. We have added this experimental approach in the manuscript as a future plan (see line 835).

You indicate that adiponectin is upregulated in the CSF (and serum) of MS patients. Can you further discuss the role of adiponectin receptor insensitivity in the mediation of MS. Based on the former, do you think that your MS-CSF protocol appropriately represents the MS ‘brain’?   Beyond cell viability, were there any other parameters that are characteristic of MS analyzed?

Reply: we thank the reviewer for the point addressed. To our knowledge, adiponectin resistance has been documented in animal models and human studies with respect to its metabolic effects. We cannot exclude, however, that MS patients also experience such a mechanism. However, the exclusion criteria included in our protocol (co-presence of metabolic conditions) impede to draw conclusions even in patients.  In addition, we are aware that our in vitro model is not fully representative of the MS brain, and therefore not suitable to deepen the adiponectin receptor insensitivity.

Our data represent a first and preliminary small step towards understanding the specific biological actions of adiponectin against MS neuronal and glial cells. Future in vivo experiments in animal models as well as MS patients with concomitant metabolic alterations would confirm our data and would clarify the adiponectin receptor insensitivity mechanism.

The effects of MS-CSF on cell viability using your cell model systems appear aggressive and immediate. Your cell data suggests that the adiponectin receptors are still sensitive to the effects of its agonists. In MS patients, where adiponectin is elevated but likely not signaling appropriately through its receptor, would Adipo-Ron have diminished efficacy? Also, can you discuss any conflicting reports where adiponectin is inversely related to MS severity.

Reply: as correctly underlined by the referee, the reduction of viability induced by MS-CSF in our cellular models is quite aggressive (between 20% and 45%) in line with previously reported data for CSF treatments (xiao et al., j neurol sci 1995; wang et al., oncotarget 2015; couratier p, et al., lancet 1993). In addition, we cannot exclude that the effects induced by AdipoRon are not entirely superimposable on those exerted by full-length native adiponectin.

The referee has underlined a critical point, considering that in vitro models we used are not totally representative of the events happening in patients. Conflicting data on the relationship between adiponectin and MS severity have been reported in literature. Accordingly, in the revised version of the manuscript we have added a landmark work (see Keyhanian et al.) where adiponectin was observed to be inversely related to MS severity (higher adiponectin levels associated with lower risk of relapse) (Mult Scler Relat Disord 2019;36:101384).

As there have been reports of sex differences in RRMS risk, why was the data not analyzed based on sex? Were there any analyzed parameters that deviated due to sex condition? Additionally, adolescent obesity status influences risk (and severity) of MS. Were there any prior history of obesity/overweight indicated for MS patients used in this study?

  • Association between comorbidity and clinical characteristics of MS
  • Influence of body mass index on psychological and functional outcomes in patients with multiple sclerosis: a cross-sectional study

Reply: as underlined by the reviewer, sex influences RRMS risk. To avoid this bias, the number of males and females from which CSF was taken is almost equal in both RRMS and PMSF patients as reported in table 1. Furthermore, we did not consider males and females separately because the clinical data of the patients divided by gender were not different (see below).

Finally, the patients who were considered for the study did not have a history of obesity/overweight.

Minor comments:

There is inconsistency in figure legend labels and what you indicate in the text. In the results section, you indicate figure 4 A, B for SH-SY5Y and U-87, respectively; but this is not the order in the actual figure. Please double check all your figures and ensure they are consistent.

Reply: we are sorry for the mistake. Accordingly, we have modified the results section.

You are missing a citation on line 216. Please check for grammatical errors and syntax (i.e., lines 195-196, lines 258 and 288). Some sentences contain weird wording (i.e., lines 260-261, 313-316).

Reply: we apologize for the mistake. We have included the reference and grammatical errors and syntax.

The statement on lines 313-314 suggest increasing sample size to possibly see differences b/w RRMS and PMS patients. In the methods section, can you indicate that your MS-CSF is pooled from n=20 RRMS and PMS subjects.

Reply: we apologize for the mistake. We now precise in the method section that the RRMS and PMS CSFs were pooled.

For the title, consider….inhibiting oxidative stress and proinflammatory  response.

Reply: Thank you for the suggestion, we have modified the title.

Round 2

Reviewer 2 Report

I have no further comments.

Author Response

We sincerely thank the reviewer for having revised our manuscript and appreciate the positive comment

Reviewer 3 Report

refer to Word doc

Author Response

Response to Author’s revisions

The revised manuscript by Mallardo and colleagues has been sufficiently improved based on reviewer’s

suggestions. The manuscript should be acceptable for publication pending resolution of the below

issues.

1. What is the n per experimental group for your MTT, NO, and LDH experiments? For gene

expression experiments, can you clarify if your sample size is n=3, or did you run triplicate

experiments with an increased number of biological replicates?

Reply: we apologize for the forgetfulness. Accordingly, we have modified the paper clarifying this point

for each experiment and figure.

Can you please indicate the sample size per treatment group. Indicating “the mean of two different

experiments” is not informative. Was there n=8 wells per treatment condition, so that your sample size

is n=16? I am assuming this based on the use of a 96-well platform (i.e., for figure 2 &3 specifically).

However, please indicate the sample size for each figure beyond figure 1, as the stats details are omitted

(i.e, critical and degree of freedom values) it is impossible to deduce.

Reply: We apologized for the inaccuracy; accordingly, we modified M&M section and figure captions in the text.

2. The asterisks on figure 4 are missing.

Reply: we apologize for the mistake. Accordingly, we modified the figure 4.

I think something has gone awry with this figure. The asterisks are still missing on all panels of the

updated figure.

Reply: Please accept our apologies. To be sure the figure is correct, we now are uploading it below and separately as an image file.

3. Figure 4- please update the figure to put a border around the NC bars in panels A and D.

Reply: Accordingly, we modified the figure as the reviewer suggested.

4. For your ANOVAs on experiments with NC, MS-CSF +/- AdipoRon, I think the analysis should not

be done using all groups as a single variable (treatment) but rather 1 factor is CSF type (diseased

or control) and the other factor as AdipoRon treatment; therefore, figures 4 and 5 should be

analyzed as 2-way ANOVAs and figure 3 as a 3-way with the factor time also assessed.

Reply: We thank the reviewer for the observation; accordingly, based on the reviewer's comments, we used MS-CSF as 1 factor and other treatments as the other factors and modified the figures in the text. However, we used one-way ANOVA for multiple comparison of the data shown in Figures 4 and 5 because we presumed treatment as the only variable and we used two-way ANOVA for the data shown in figure 3 because we presumed time and treatment as variables; we modified the figure by comparing treatments versus MS-CSF.

If we done something wrong in the analysis we are ready to modify according to the reviewer’s comments.

5. Please make the following corrections (add underlined words to replace or omit italicized

terms): (i.e., lines 27 & 29 (whereas it determines, while it determines), 32 (replace one of the in

vivo with in vitro, 63 (replace crass with cross talk), 65 (remove the word other), 68 (correct to

participating in the bidirectional cross talk), 85 (replace outiline with outline), 114 (EDSS), and

total), 195 (Statis ??), 205 (DMEM supplemented with 5%), 208 (replace trought with

throughout), 249 (its potential in interfering with MS-CSF-induced…), 251 (replace reduces with

reduced), 252 (SH-SY5Y (a) and U-87 (c), 254 (SH-SY5Y (b), 325 (remove the word some), 339 (as

a prognostic...), 363 (remove thee), 367 (levels whereas it caused the), 379 (replace counteraton

with counteraction), 402 (consider: Nonetheless, altogether our data…)

6. In the title for Cerebrospinal fluid- use lowercase ‘c

Reply: we sincerely thank the reviewer for his/her comments and corrections and for having revised the text.